

# Evaluation model design of project construction safety level based on bidirectional recurrent neural network (BiRNN) and bidirectional long short-term memory (BiLSTM)

Ming Ge and Yongbo Yuan

School of Infrastructure Engineering, Dalian University of Technology, Dalian, Liaoning, China

## ABSTRACT

Integrating deep learning methods for multi-element regression analysis poses a challenge in constructing safety evaluations for building construction. To address this challenge, this paper evaluates the integration of construction safety by quantitatively analyzing practitioners' information and on-site construction conditions. The analytic hierarchy process (AHP) method quantifies construction safety capabilities, considering four key aspects: operators' primary conditions, organizational personnel's working conditions, on-site management conditions, and analysis of unsafe behaviors. A comprehensive set of 19 secondary causal factors is constructed. Furthermore, a hybrid model based on bidirectional recurrent neural network (BiRNN) and bidirectional long short-term memory (BiLSTM) is developed for construction safety evaluation, enhancing the model's generalization ability by introducing the Dropout mechanism. Experimental results demonstrate that the fusion of BiRNN and BiLSTM methods outperforms traditional methods in construction safety evaluation, yielding mean squared error (MSE) and root mean squared error (RMSE) values of 0.48 and 0.69 and mean absolute error (MAE) and mean absolute percentage error (MAPE) values of 0.54 and 3.36%, respectively. The case study affirms that BiRNN-BiLSTM can accurately identify potential safety risks, providing reliable decision support for project management.

**Submitted** 2 July 2024
**Accepted** 2 September 2024
**Published** 18 October 2024

Corresponding author
Ming Ge, geming@mail.dlut.edu.cn

## INTRODUCTION

The construction industry is one of the most perilous sectors globally, inflicting substantial casualties and property losses across countries. With the ongoing refinement of building construction production technology and the amplification of the scale and intricacy of infrastructure projects, the imperative for safety performance and the management of safety aspects in engineering construction has progressively evolved into a paramount challenge for the entire industry (*Akinlolu et al., 2022*; *Zhou et al., 2023*). The construction of buildings is a multifaceted and high-risk systematic endeavor, distinguished by its extensive construction scale, numerous participants, intricate technical processes, and

fluctuating operational environments. Diverse unforeseeable safety risk factors permeate the processes of various building constructions, posing a considerable likelihood of precipitating safety incidents.

Construction engineering manifests as a temporally dynamic system characterized by intricate phenomena and chaos (*Asadzadeh et al., 2020*). From the vantage point of safety system engineering, the safety state of a project emerges as a complex adaptive system, engaging continuously with the external environment and perpetually adjusting its internal structure and behavior. Historically, myriad experts and scholars undertook risk identification, assessment, and response for subway construction using accumulated historical data and experiences from building construction. Nevertheless, owing to the intricate geological and hydrological environment, diverse objects, technical complexities in construction, and organizational and coordination challenges in construction, the safety risks adopt complex, concealed, and dynamic characteristics. Even the most seasoned safety experts find it challenging to analyze and preemptively discern all safety conditions at the intricate and mutable project site. Cognitive bias and individual subjectivity inevitably impede their judgments (*Su et al., 2021*). As the historical data accrues continuously, the dataset becomes enriched, posing substantial difficulties in comprehensive analysis due to the typical spatial and temporal characteristics, highly nonlinear features, and intricate coupling effects (*Mostofi & Toğan, 2023*; *Ajayi et al., 2020*).

Building upon the research outcomes of experts in the field, a contingency exists to explore safety risk management technology founded on data mining, data analysis, and data-driven safety risk management technology. This entails utilizing information technology to sift through more significant volumes of data, encompassing historical data across diverse accident types, for the automated identification of safety risks. Building upon the results of data analysis, industry personnel are assisted in conducting assessments of safety risks and fortifying the safety capabilities in enterprise operations. This endeavor propels enterprise safety management, fostering the realization of the company's benign development.

Traditionally, an enterprise's construction safety proficiency is typically ascertained through considerations such as personnel structure, performance metrics, customer evaluations, historical assessments, and other pertinent factors (*Singh & Misra, 2021*). While this approach can assess the overall safety level of enterprise construction to a considerable extent, the data support needs more scientific precision and meticulousness, with a less precise and intuitive degree of quantification. The analysis content, too, appears relatively coarse, necessitating further refinement and enhancement. Project quality is influenced by myriad factors, predominantly encompassing people, materials, equipment, methods, and environmental conditions. Among these factors, human elements are the foremost and most pivotal. Personnel are the principal agents in executing behaviors throughout the construction project, exerting the most substantial impact on project management, organization, guidance, operations, and other skills integral to project quality (*Singh & Misra, 2021*). For instance, inadequate use of safety belts and other safety equipment by workers engaged in elevated work introduces an elevated risk of falls.

Moreover, proper safety training may lead workers to properly deploy tools and equipment, resulting in incorrect usage and subsequent safety incidents.

Construction safety risks' intricate and uncertain nature defies accurate representation through specific mathematical or algorithmic models. With the ascension and evolution of extensive data analysis methods centering on artificial neural network algorithms, the feasibility of risk prediction based on quantitative data from historical risk assessments has emerged in recent years. However, prevailing efforts primarily revolve around neural networks such as back propagation (BP) and radial base function (RBF), which need help to align with the characteristics of quantitative data from historical risk assessments. Shortcomings like local convergence and an inability to effectively support long-time series data persist (*Shen, Nagai & Gao, 2020*; *Zhou et al., 2023*; *Alkaissy et al., 2020*). In a bidirectional recurrent neural network (BiRNN), the forward pass processes the data from the beginning to the end of the sequence, capturing temporal dependencies that inform the model about past events. Simultaneously, the backward pass processes the data from the end to the beginning of the sequence, incorporating information about future events. This dual-pass mechanism enables BiRNNs to build a more comprehensive context, making them particularly adept at handling complex, long-term dependencies in time series data. Consequently, this paper endeavors to investigate the impact of an enterprise's personnel reserve, personnel composition, personnel input (carrying), fundamental security, and other factors on the safety and stability of the construction site through modeling and computational analysis. An objective comparison between the modeling and computational analysis results and the enterprise's safety performance reveals a fundamental trend consistency. This approach aims to evaluate the safety landscape within the regional construction force comprehensively. Following the quantitative analysis of employee information and on-site construction conditions, construction safety assessment and integration are undertaken across multiple abstract dimensions.

In this study, we apply BiRNN and bidirectional long short-term memory (BiLSTM) algorithms to model and analyze the impact of factors such as personnel reserve, composition, input, and basic security on construction site safety and stability. By integrating and training these models with detailed quantitative data, we aim to evaluate the safety landscape within the regional construction workforce comprehensively. The results from our modeling and computational analysis will be compared with actual safety performance metrics to validate the effectiveness and accuracy of the proposed approach.

As construction safety risk assessment is critical for ensuring project sustainability, this paper also addresses the challenge of systematically collecting and integrating a comprehensive range of accident risk factors from real construction sites. Through rigorous model training and optimization, this research seeks to provide a detailed, quantitative assessment of construction site safety, offering valuable insights into risk management and mitigation strategies.

## RELATED WORKS

Safety risk assessment hinges on identifying safety risks and, contingent upon the potential severity of accidents, calculating and determining the risk level of the identified factors.

Qualitative methodologies commonly employed encompass the Fault Tree Analysis (*Bakeli & Hafidi, 2020*), the Comprehensive Fuzzy Evaluation Method (*Guo & Wu, 2023*), and the Construction Safety Checklist (*Park et al., 2021*). Quantitative approaches include artificial neural network (ANN) (*Shen, Nagai & Gao, 2020*), support vector machine (SVM) (*Liu et al., 2020*), Bayesian network (BN) (*Fang et al., 2023*), and Decision Tree (*Zhu et al., 2021*). *Liang & Liu (2022)* introduced a risk-based safety impact evaluation method for underground engineering, utilizing example analysis and survey methodologies, encompassing safety information surveys, classification of safety impact factors during design and construction, and quantitative estimation of magnitude and frequency. *Lin et al. (2021)* amalgamated weight dynamic adjustment, fuzzy comprehensive judgment, and logical layer risk evaluation methods, proposing a systematic approach applicable to the construction safety risk evaluation of large-scale foundation pits situated in proximity to existing buildings. *Wang et al. (2023)* devised a comprehensive decision support method for uncertain safety risk analysis in tunnel construction, employing Bayesian network and fuzzy set theory. This extended the analysis process to encompass the entire life cycle of construction risk events, spanning continuous control before the accident, during construction, and post-accident. *Cao, Li & Hou (2022)* utilized simulation to scrutinize a subway system's typical scale-free network characteristics, showcasing high robustness in the face of random faults and low fault tolerance against malicious attacks. *Luo et al. (2022)* integrated the analytic hierarchy process (AHP) and entropy weights to determine weights. They established a multilevel topological evaluation model for the safety risk of subway construction, emphasizing the topological method.

The AHP offers a structured and objective approach to decision-making that is particularly valuable in safety evaluations. By organizing complex decisions into a hierarchical structure, AHP facilitates the systematic assessment of multiple criteria and alternatives. This structured approach helps mitigate the subjectivity inherent in expert judgments by quantifying subjective opinions through pairwise comparisons and deriving consistent weightings. Consequently, AHP improves the objectivity and reproducibility of safety evaluations. Moreover, AHP enhances risk prioritization by evaluating the relative importance of various safety factors within a unified framework, ensuring critical risks are identified and addressed effectively.

Traditional automated risk identification relies on patterns, constraints, and machine learning techniques to extract safety risk factors and their causal relationships. This approach heavily depends on domain knowledge and demands significant human resources and time for feature engineering, resulting in notable limitations (*Qayyum et al., 2020*). Complex network-based security risk management analyses often concentrate on independent risk factors or a singular risk event type, with limited exploration of intricate interactions between security risk factors and diverse risk event types. Moreover, addressing the challenge of eliminating cognitive bias and individual subjectivity in risk analysis based on expert experiences proves difficult.

Traditional construction safety evaluation methods face constraints imposed by statistical models and regularity methods, impeding the full exploitation of complex time-series data features. In recent years, the emergence of deep learning technology has

ushered in new solutions for construction safety evaluation. *Zhao et al. (2023)* explored the application of convolutional neural networks (CNN) for construction safety event prediction but identified limitations in capturing temporal relationships. Numerous studies have leveraged recurrent neural networks (RNN) or RNN variants like long short-term memory (LSTM). *Guo et al. (2022)* utilized Bi-LSTM to predict safety levels in subway construction, achieving an average absolute error of 13% on the test dataset.

In recent advancements, the application of deep learning in construction safety has seen substantial progress, addressing some of the limitations of earlier approaches. *Xiang et al. (2024)* explored the use of Transformer models for safety risk prediction, demonstrating improved performance in handling long-term dependencies and complex interactions between risk factors. Their study showed that the Transformer's self-attention mechanism outperformed traditional RNN-based models regarding accuracy and interpretability. Additionally, *Thakur, Kansal & Rishiwal (2024)* proposed a hybrid deep learning framework combining CNN and LSTM networks for real-time safety hazard detection. Their approach leverages CNNs for spatial feature extraction and LSTMs for temporal sequence modeling, significantly enhancing real-time safety monitoring and hazard prediction accuracy.

*Li et al. (2021)* introduced a short-term load forecasting method with LSTM-RNN considering the energy storage effect, demonstrating MAPE of 3.0 and RMSE of 0.72. *Zheng et al. (2022)* proposed a multiscale RNN-based method with high precision, showcasing potential as a robust solution for output load prediction. *Zhu & Wang (2021)* employed feature selection and an LSTM model to analyze road construction safety conditions, revealing improved prediction results with LSTM. *Pham et al. (2021)* introduced a construction safety evaluation model based on RNN, outperforming traditional safety event identification and risk assessment methods. *Deng et al. (2020)* incorporated BiLSTM to address long-term dependencies, yielding strong performance in time-series data. Compared to traditional models, comparative experiments showcased higher accuracy and robustness in construction safety event prediction with BiLSTM. *Yang, Zhang & Ai (2024)* conducted an in-depth comparative study on BiRNN and BiLSTM, highlighting BiRNN's suitability for dynamic time-series data and BiLSTM's proficiency in handling long time-series dependencies. Thus, choosing appropriate models according to specific construction scenarios enhances the evaluation system's performance.

The hybrid BiRNN-BiLSTM model is well-suited for handling sequential and time-series safety data due to its advanced capabilities in capturing temporal dependencies and contextual information. The BiRNN and BiLSTM models excel in processing sequential data, making them ideal for analyzing historical and real-time safety data. The BiRNN component allows the model to consider information from both past and future contexts, while the BiLSTM component manages long-term dependencies and mitigates issues such as vanishing gradients. This combination enhances prediction accuracy and adaptability to evolving safety threats. The model's ability to continuously learn from sequential data ensures that it remains responsive to changes in the safety environment, making it a valuable tool for improving safety evaluations.

In contrast to these recent studies, our work introduces a novel approach by integrating advanced deep learning techniques with a multi-modal data fusion strategy. While previous studies have primarily focused on single data types or specific deep learning architectures, our approach combines diverse data sources, such as real-time sensor data, historical safety records, and contextual information, through a unified deep learning framework. This integration enhances the ability to capture complex interactions and dependencies across multiple dimensions of construction safety data, leading to more accurate and actionable risk predictions. Moreover, our study addresses the challenge of cognitive bias and subjectivity in risk assessment by incorporating automated feature extraction and data-driven decision-making processes, reducing reliance on domain expertise and manual intervention. This advancement represents a significant improvement over existing models, providing a more objective and scalable solution for construction safety evaluation.

## METHODOLOGY

### Construction of evaluation indicators

Enforcing a real-name system for workers and establishing authenticated access prevents unauthorized individuals from entering the site. This enhances internal safety management, ensuring project quality and safe production and providing valuable information support for industry policy development. Implementing a real-name system for pre-job training and assessment aids the owner in worker selection. Utilizing the AHP, 19 secondary causal factors are constructed based on the primary condition of operating personnel, the working conditions of organizational and management personnel, on-site management conditions, and the analysis of unsafe behaviors, as depicted in Table 1.

In this study, exclusive attention is directed towards target layer A and criterion layer B. The focus is solely on unraveling the relationship between the target layer (A) and the criterion layer (B), specifically solving for the criterion layer B in relation to the target layer A. Determining weights assigned to the criterion layer concerning the target layer is undertaken. The elements of the B-layer are delineated using a rating scale ranging from 1 to 4 and its reciprocal. Subsequently, a judgment matrix is formulated to portray the relative importance of each element.

$$B = \{b_{ij} | i, j = 1 \sim n\}. \tag{1}$$

Assuming B be the single-ordered weights of the layers are $w_k, k = 1 \sim n$, and satisfy $w_k > 0$ and $\sum_{k=1}^{n} w_k = 1$, according to the judgment matrix, Eqs. (2) and (3) should be satisfied.

$$b_{ij} = w_i / w_j, i, j = 1 \sim n \tag{2}$$

$$\left| \sum_{k=1}^{n} (b_{ik} w_k) - n w_i \right| = 0. \tag{3}$$

In the resolution process, obtaining eigenvectors that precisely adhere to theoretical requirements poses challenges. Leveraging the principles of hierarchical analysis reveals that

**Table 1  Construction safety evaluation indicators.**

| Hierarchy of causes | Contributing factor |
|---|---|
| Operator a1 | Age b11 |
| | Educational attainment b12 |
| | Acquisition of qualifications b13 |
| | Years of service b14 |
| | Electrical work b15 |
| | Quality assurance assignment b16 |
| | Construction excavation operations b17 |
| Site management a2 | Real name registration and attendance b21 |
| | Pre-construction safety technical communication b22 |
| | Construction crew site location b23 |
| | Construction personnel working hours b24 |
| Organizational managers a3 | Education and training b31 |
| | Safety supervision and inspection, acceptance b32 |
| | Risk monitoring b33 |
| | Emergency relief b34 |
| | Accident reporting, investigation and handling b35 |
| Unsafe behavior a4 | Perception and decision errors b41 |
| | Skill error b42 |
| | Operational violation b43 |

matrix consistency signifies the quality of eigenvectors (*Pant et al., 2022*). Consequently, the task of determining eigenvectors can be transformed into the endeavor of resolving the optimal consistency value.

$$\min \text{CI} = \sum_{i=1}^{n} \left| \sum_{k=1}^{n} (b_{ik} w_k) - n w_i \right| / n$$

$$\text{s.t.} \quad w_k > 0, k = 1 \sim n \quad (4)$$

$$\sum_{k=1}^{n} w_k = 1.$$

According to the definition of consistency ratio, when $\text{CR} = \frac{\text{CI}}{\text{RI}} < 0.1$, the judgment matrix is considered to have satisfactory consistency.

## BiRNN

The complete sequence of features undergoes encoding using a BiRNN network, which records the context vector for each output vector. The current state is forecasted by the forward RNN, capturing the before-mentioned relationships, while an inverse RNN predicts the character based on the context. Ultimately, the prediction results from both RNNs are linearly combined to generate the output of the final bidirectional RNN.

The forward network of the BiRNN iterates from time step $t = 1$ to T and the backward network iterates from time step $t = T$ to 1. This involves simultaneous computation of the forward hidden sequence $h_f$ and the backward hidden sequence $h_b$. The two sequences are
summed to update the output sequence y, following the specifications in Eqs. (5), (6) and (7) (*Hernandez-Matamoros, Fujita & Perez-Meana, 2020*).

$$h_f = W_{xh_f}x_t + W_{h_f}h_{f(t-1)} + b_{h_f} \tag{5}$$

$$h_b = W_{xh_b}x_t + W_{h_b}h_{b(t-1)} + b_{h_b} \tag{6}$$

where $W_{xh_f}$ and $W_{xh_b}$ represent the weight matrices that transform the input $x_t$ into the hidden state space for the forward and backward networks, respectively. $W_{h_f}$ and $W_{h_b}$ are weight matrices that connect the hidden states from the previous time step to the current hidden state. The bias vectors $b_{h_f}$ and $b_{h_b}$ are added to the hidden states to allow the model to learn an offset for better fitting the data.

At each time step t, the forward and backward hidden states $h_{ft}$, and $h_{bt}$ are combined to produce the output sequence $y_t$ as follows.

$$y_t = W_{h_f}h_{ft} + W_{h_b}h_{bt} + b_y. \tag{7}$$

$Wh_f$ and $W_{h_b}$ are weight matrices that project the forward and backward hidden states into the output space, and $b_y$ isthe bias vector applied to the output. The final output $y_t$ is thus a weighted combination of the forward and backward hidden states, adjusted by the bias.

## BiLSTM

Various design structures exist for the state memory function of sequences. LSTM, employed in this study, represents one such design for RNN units. LSTM captures connections between sequence contexts, learns probabilistic relationships between contexts, and finds extensive applications in natural language processing (*Su & Kuo, 2019*). Its unique ability to regulate information flow is achieved by introducing a gate structure, automatically determining whether information should be retained or discarded. This design also addresses the challenges of gradient explosion and gradient vanishing encountered during the training of RNN networks.

BiLSTM, an extension of LSTM, enhances the model's capacity to consider historical and future information when processing sequence data. While standard LSTM processes input sequence information from left to right, BI-LSTM concurrently processes left-to-right and right-to-left information (*Wu et al., 2023*). The computational procedure of BiLSTM is outlined in Eqs. (8) and (9):

$$\vec{h}_t = \overrightarrow{\text{LSTM}}(x_t, h_{t-1}) = \sigma\left(W \cdot [h_{t-1}, x_t] + b\right) \tag{8}$$

$$\overleftarrow{h}_t = \overleftarrow{\text{LSTM}}\left(x_t, \overleftarrow{h}_{t+1}\right) = \sigma\left(W \cdot \left[\overleftarrow{h}_{t+1}, x_t\right] + b\right) \tag{9}$$

where $h_t$ is the time step $t$ of the hidden state, $\vec{h}_t$ is the output of forward LSTM, and $\overleftarrow{h}_t$ is the output of the inverse LSTM. The two merged outputs in the sequence process are shown in Fig. 1:

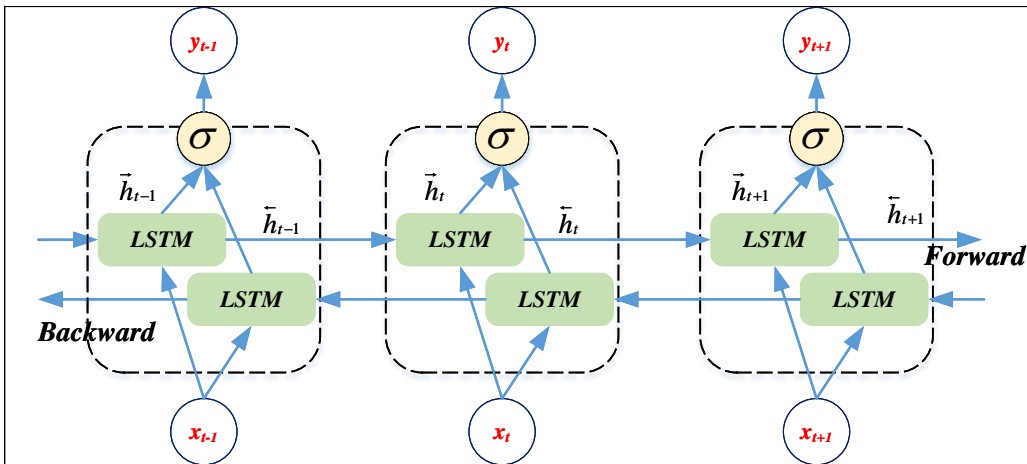

**Figure 1** The BiLSTM network framework.

## Model construction

This study employs a sophisticated model architecture integrating BiLSTM and BiRNN layers to enhance prediction accuracy. The process begins with data preparation, where all input features and labels are normalized based on the training dataset. This normalization is crucial for reducing the influence of individual sample data and ensuring consistent scaling across the model, which ultimately contributes to more reliable predictions.

Following data normalization, the processed data is input into the RNN layer. The RNN layer's output undergoes regularization through a Dropout layer to address the overfitting issue. This layer randomly drops units during training, which helps prevent the model from becoming overly dependent on specific nodes and improves generalization. The regularized output is then forwarded to the LSTM layer, which similarly applies Dropout to its output to maintain regularization throughout the network.

Subsequently, the output from the Dropout layer is passed to the BiLSTM layer. The BiLSTM layer processes the data in both forward and backward directions, more comprehensively capturing temporal dependencies. The data then progresses to the BiRNN layer, further enhancing the model's ability to learn from sequential information by processing data in both directions.

The final stage consolidates the predictions through a fully nonlinear superposition, integrating the outputs from the BiLSTM and BiRNN layers. This consolidated output is then processed through a fully connected Dense layer, which generates the final prediction results. After training, the model's parameters are saved, and test data is normalized and input into the model. The predictions obtained from the model are subsequently inverse-normalized to produce the final outcomes.

This multi-layered approach thoroughly integrates recurrent layers, enhancing the model's predictive performance. Carefully structuring these layers and applying regularization techniques collectively improve the model's ability to generalize and deliver accurate predictions.

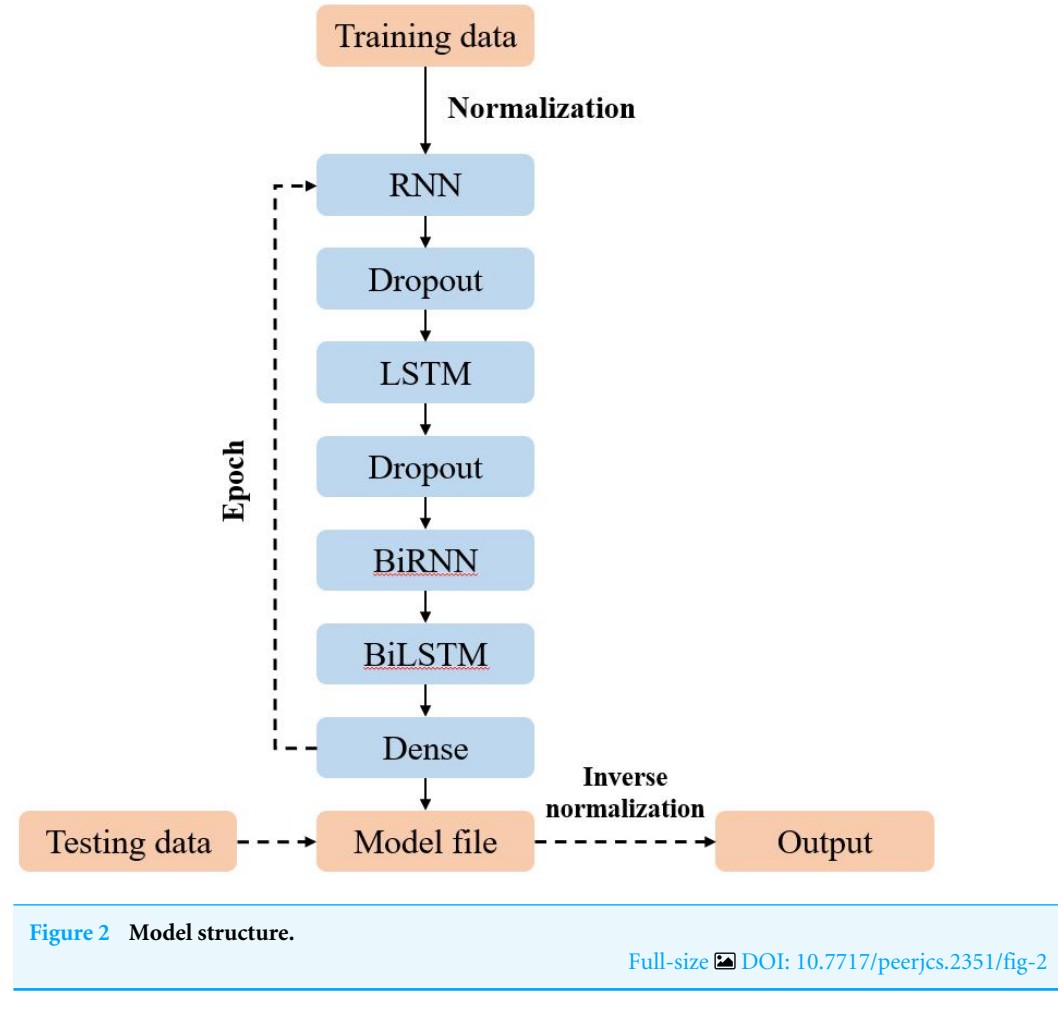

**Figure 2   Model structure.**

The process of constructing the model is depicted in Fig. 2.

## EXPERIMENTATION AND ANALYSIS

### Data sources

This study gathers real-name list information for over 1,000 construction and organization management personnel from over 40 enterprises (https://zenodo.org/records/7930450, doi: 10.5281/zenodo.7930450). To enhance the effectiveness of the real-name system, construction enterprises are encouraged to adopt information-based management approaches. This comprehensive approach aims to improve efficiency across all facets of the real-name system, encompassing contract signing, work attendance, payroll management, safety protocols, and skills training.

### Experimental setup

The algorithm simulation is conducted within the computer hardware and software environment, as detailed in Table 2.

The algorithm operation parameters are set as shown in Table 3.

**Algorithm Framework**

1. **Load Data**: Load the input data and corresponding labels.
2. **Data Normalization**: Normalize both input data and labels to ensure uniform scale.
3. **Model Construction**.
   - Construct a sequential model.
   - Add a SimpleRNN layer with 32 units and an input shape derived from the data.
   - Apply dropout with a rate of 0.35 to mitigate overfitting.
   - Add an LSTM layer with 64 units.
   - Apply dropout again to the LSTM layer.
   - Add a Bidirectional LSTM layer with 128 units.
   - Include a Dense layer with 1 unit for the final output.
4. **Model Training**.
   - Compile the model using the Adam optimizer with a learning rate of 0.01.
   - Train the model for 200 epochs with a batch size of 25.
5. **Load Test Data**: Load the test dataset.
6. **Data Normalization for Test Set**: Normalize the test data using the same normalization parameters as the training set.
7. **Model Prediction**: Use the trained model to predict the test set.
8. **Inverse Normalization**: Reverse the normalization process to obtain the final predictions.
9. **Results**: Display the final predictions.

**Table 2  Parameters of simulation environment.**

| Project | Parameter |
| --- | --- |
| Operating system | Windows 10 |
| Cpu | Intel core i7-11700 |
| Random access memory (RAM) | kf432c16bbk2/16-sp |
| Display card (computer) | GTX 1050 |
| Programming environment | Python 3.5 |

**Table 3  Model parameter settings.**

| Parameter name | Parameter value |
| --- | --- |
| Dropout probability | 0.35 |
| Initial learning rate | 0.01 |
| Termination rate | 0.1 |
| Iterations | 200 |
| Batch size | 25 |
| Number of training set | 800 |
| Number of testing set | 200 |

## Training results

The model in this study conducts crisis prediction on the training set, yielding the confusion matrix illustrated in Table 4.

**Table 4  Confusion matrix of model training.**

| Categorization | Results | | Number of projects | Accuracy/% |
|---|---|---|---|---|
| | Construction safety hazards exist | No construction safety hazards | | |
| Construction safety hazards exist | 29 | 2 | 31 | 96.55% |
| No construction safety hazards | 3 | 29 | 32 | 93.6% |

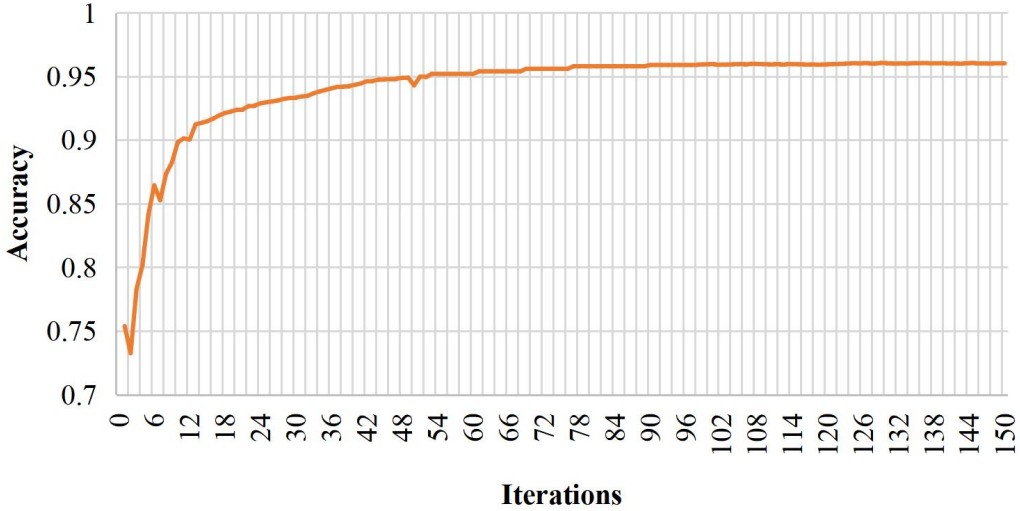

**Figure 3  Model training diagram.**

Analyzing the prediction outcomes of the training samples reveals two misclassifications out of 31 positive samples and three discrimination errors in 32 negative samples, totaling five classification errors across 63 samples. Consequently, the deduced accuracy of the model in this study is 95.24%, and the Recall is 96.55%. Figure 3 depicts the classification accuracy curve of the BiRNN-BiLSTM model at 150 iterations on the training set.

The model in this paper demonstrates the robustness and generalization capabilities by converging and stabilizing after 18 iterations. The convergence at an early stage signifies efficient learning and optimal performance on the given training set. This outcome highlights the model's adaptability to different data instances, showcasing its ability to generalize effectively beyond the training data. The convergence and stability at a relatively low iteration count also suggest that the model successfully captures the underlying patterns and relationships within the dataset, providing confidence in its reliability for real-world applications and predictive tasks.

## Model comparison

Utilizing prediction accuracy as the evaluation metric, the prediction results of different models are depicted in Fig. 4. Notably, the traditional LSTM network exhibits a 4.59% improvement in accuracy compared to RNN, showcasing the enhanced performance of

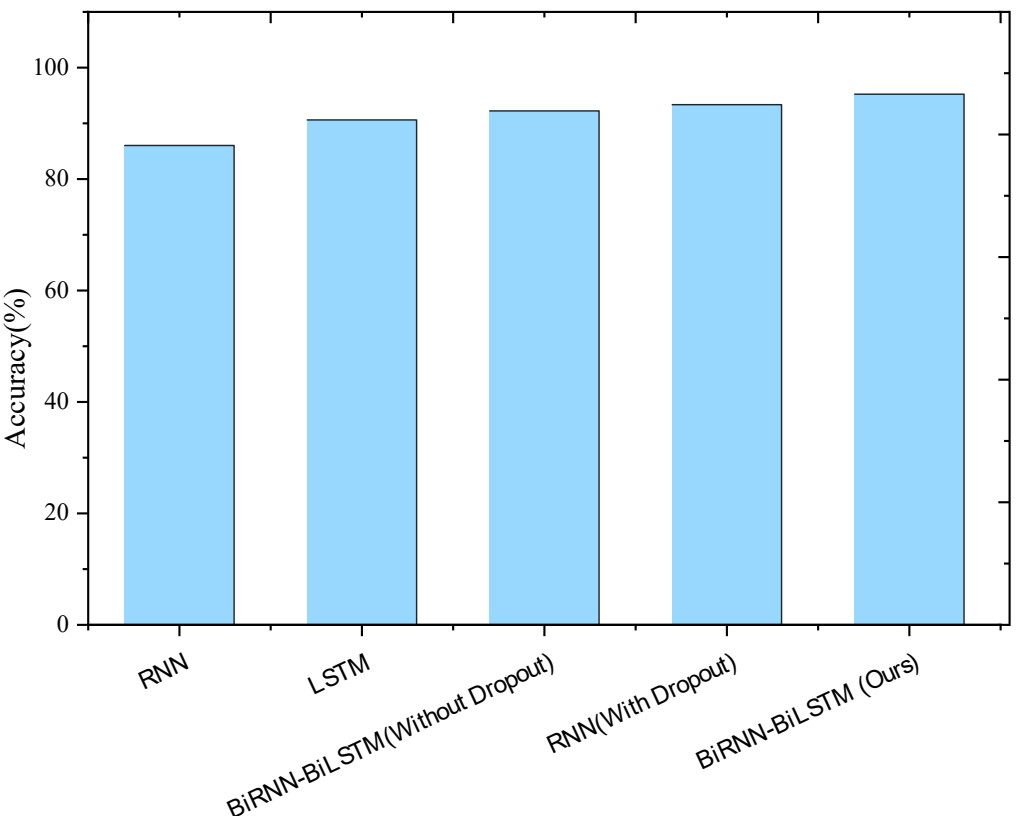

**Figure 4** **Results of ablation experiments.**

LSTM networks in analyzing and predicting time series. Additionally, introducing BiLSTM cells and the Dropout mechanism further improves the accuracy of the RNN network in model prediction. Specifically, the prediction accuracy of the BiLSTM model increases by 1.60%, the introduction of Dropout improves accuracy by 2.72%, and incorporating both mechanisms simultaneously enhances the prediction accuracy by 4.61%. This underscores the positive impact of these enhancements on the model's predictive capabilities. Furthermore, the Attention-based Time-Incremental Network (ATIN), RNN-LSTM, BiGRU, and TCN are included to compare with the model proposed in this study. The results are illustrated in Figs. 5 and 6.

BiRNN-BiLSTM (Ours) demonstrates superior performance in terms of mean squared error (MSE) and root mean squared error (RMSE) with values of 0.48 and 0.69, respectively. This signifies a significant advantage in minimizing squared errors and efficiently capturing the difference between predicted and actual values. The ATIN model also displays favorable results with relatively low MSE (0.51) and RMSE (0.71), indicating good performance regarding both squared and relative magnitude of the error. Conversely, the BiGRU model exhibits higher MSE (0.78) and RMSE (0.88), suggesting comparatively poorer performance in error minimization.

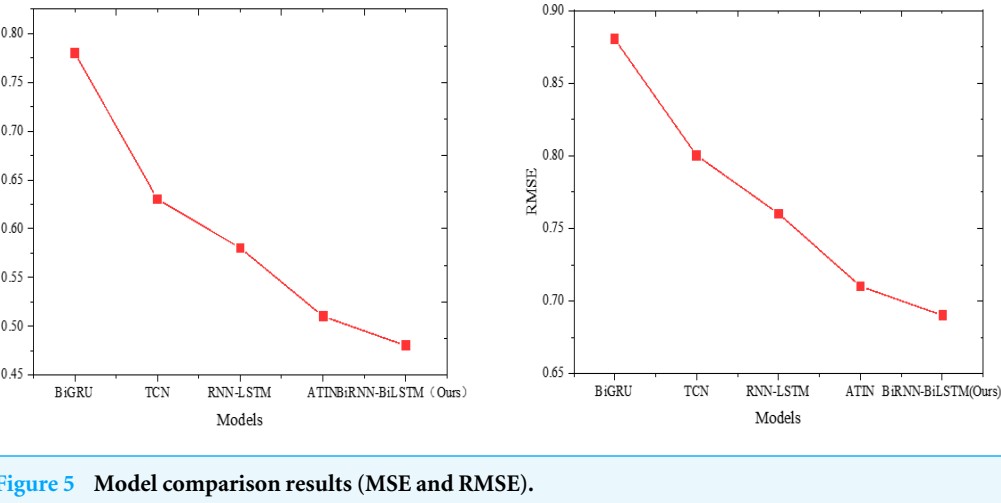

**Figure 5** Model comparison results (MSE and RMSE).

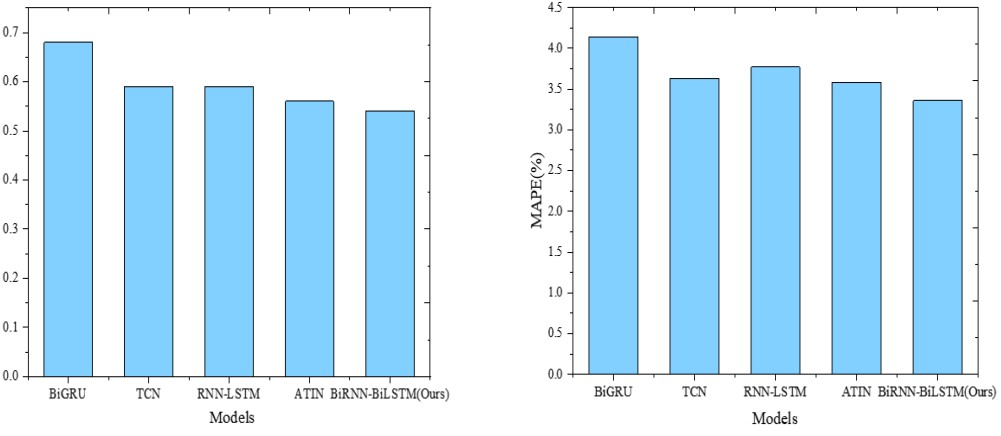

**Figure 6** Model comparison results (MAE and MAPE).

Our focus centers on the model's capacity to capture the absolute and relative percentages of prediction error accurately. In this context, BiRNN-BiLSTM (Ours) once again excels, demonstrating the lowest mean absolute error (MAE) at 0.54 and the lowest mean absolute percentage error (MAPE) at 3.36%. The ATIN model also performs well, yielding a low MAE of 0.56 and MAPE of 3.58%. Conversely, the BiGRU model exhibits a higher MAE of 0.68 and a higher MAPE of 4.14%, indicating relatively weaker performance.

The outstanding performance of BiRNN-BiLSTM (Ours) can be attributed to the advantageous features embedded in its model architecture. The combination of BiRNN and BiLSTM enables the model to capture long-term dependencies more effectively in time-series data. This bidirectional recursive structure enhances the model's understanding of contextual information within sequences during training, improving its ability to learn complex patterns in time-series data. Additionally, the bidirectional structure facilitates a

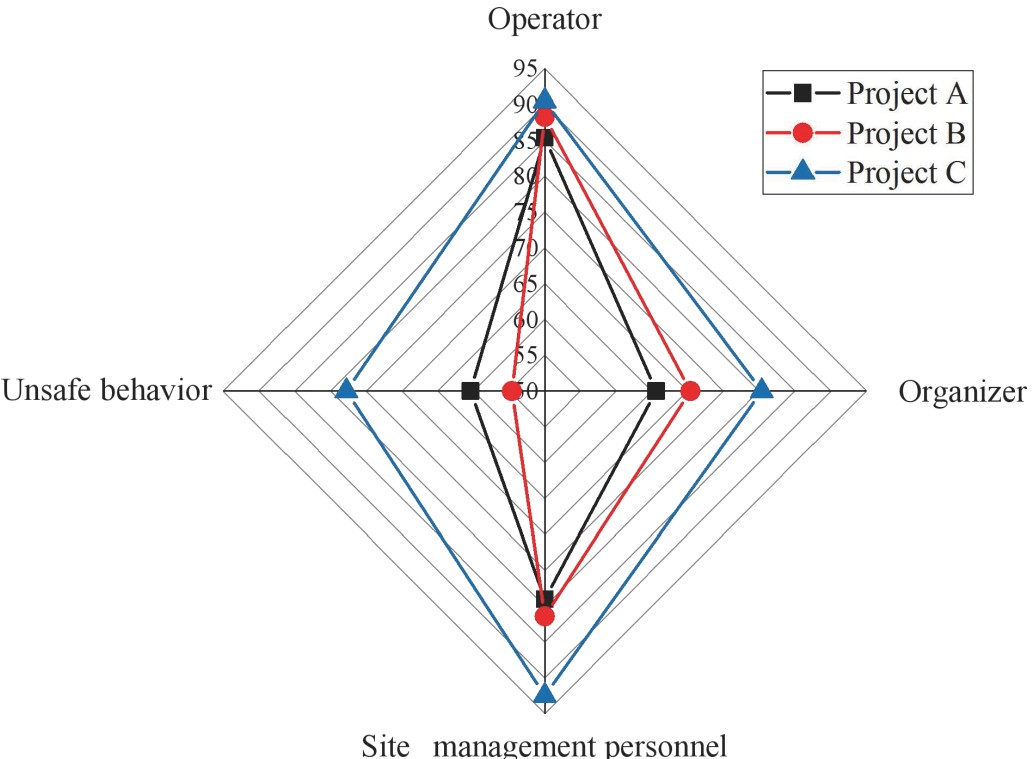

**Figure 7** **Radar charts for first level indicator scores.**

more comprehensive utilization of historical information, rendering the model adaptive and robust.

## Case analysis

The study focuses on three construction projects—Project A, Project B, and Project C—situated in the power industry sector in the Qingpu District of Shanghai. These projects were assessed based on their safety performance, which was categorized into five distinct levels: $V = [V1, V2, V3, V4, V5] = [(0,30), (30,60), (60,70), (70,80), (90,100)]$. These levels correspond to "very poor", "poor", "average", "good", and "excellent", respectively.

The data collection process involved several systematic steps to ensure accuracy and comprehensiveness. Initially, safety performance data were systematically gathered from each of the three construction projects. This data collection included direct observations, safety reports, and incident records. Observations were conducted on-site to capture real-time safety practices and compliance with safety regulations. Safety reports provided detailed accounts of safety performance over time, while incident records offered insights into any safety breaches or accidents that occurred. The results are depicted in Fig. 7.

Figure 7 shows that Project C outperforms Project A and Project B in overall construction safety evaluation. Project B lags, particularly regarding unsafe behaviors, while Project A exhibits a lower evaluation value in on-site management, creating a discernible gap with Project C. The variations in scale and composition among these projects

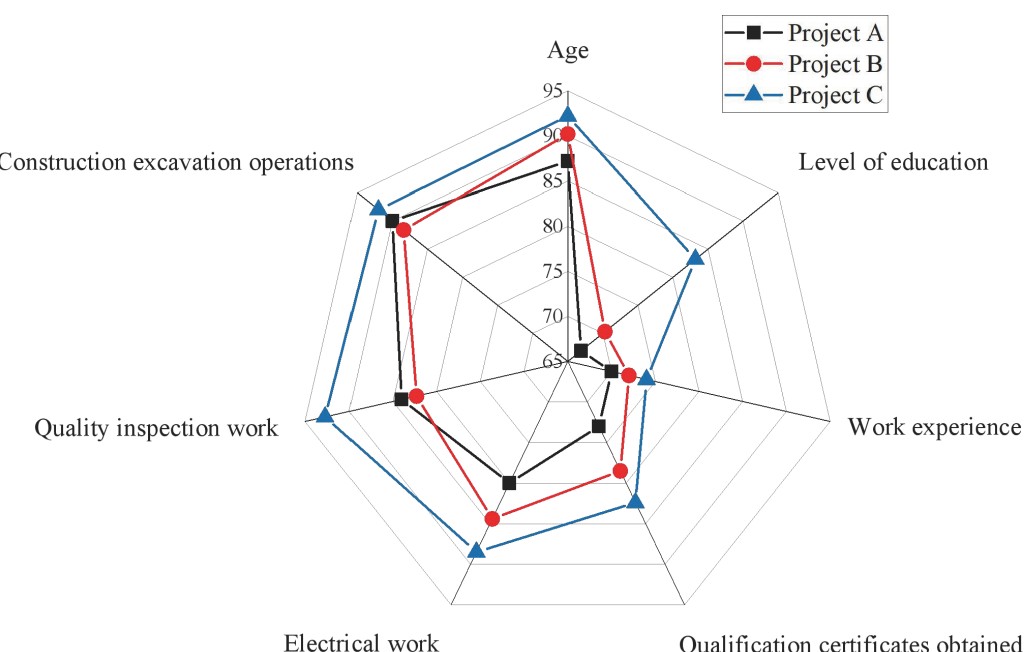

**Figure 8** **Radar chart of secondary indicator scores.**

contribute to differences in their construction safety evaluation levels. These distinctions underscore the efficacy of this study's index system and model as an effective measure for comprehensive construction safety evaluation. Despite these differences, the three selected projects consistently receive high scores in evaluating operating personnel, attributed to implementing the real-name system and ensuring information security management and unity among engineering personnel. The analysis primarily focuses on the basic situation of the operating personnel, as illustrated in Fig. 8.

Firstly, examining the data on operational staff performance reveals that Project C outshines the others with a score of 90.5, significantly surpassing Project A (85.4) and Project B (88.3). Secondly, regarding site management, Project C demonstrates superior operational and management efficiency with a score of 80.4. In contrast, Project A (65.6) and Project B (70.4) lag, suggesting that Project C exhibits higher organizational coordination and site management. At the organizational management personnel level, Project C again excels with a score of 92.5, markedly higher than Project A (79) and Project B (81.4), indicating superior leadership and organizational coordination capabilities. However, the data on unsafe behaviors reveal that Project C has a relatively high score (77.8), potentially signifying a higher frequency of unsafe behaviors. In contrast, Project A (60.4) and Project B (54.6) score lower, indicating better safety practices. This aspect requires attention, prompting a more in-depth safety assessment and implementing improvement measures.

Figure 9 presents the predicted construction safety evaluation scores of Projects A, B, and C. The scores show that Project C's construction safety evaluation score gradually improves as the project progresses. In contrast, Projects A and B experienced a decline,

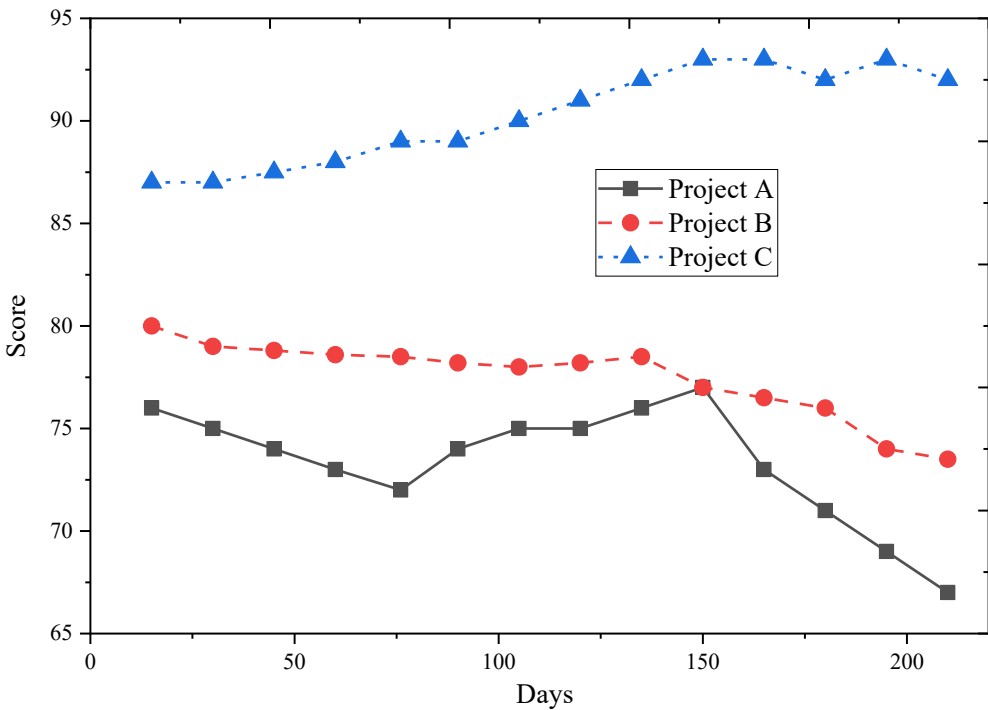

**Figure 9** **Predicted construction safety evaluation score.**

with Project A showing a noticeable decrease, indicating potential safety hazards in the later stages of the project.

Considering the results from Figs. 7 and 8, it is evident that Project C excels in organizational management, which primarily contributes to its higher construction safety evaluation score. The outcomes of this assessment demonstrate the model's accuracy in analyzing the dynamic safety evaluation scores of various project sites. This insight is valuable for managers, providing guidance to facilitate timely rectification and effective management of project sites.

## DISCUSSION

Regarding model validation, the improved performance of BiRNN-BiLSTM can be attributed to its ability to handle outliers and volatility effectively. The bi-directional structure allows the model to efficiently capture sudden and anomalous sequence variations, enhancing its stability. This stability contributes to more accurate measurement of prediction errors and reduces the fluctuation of absolute errors. Additionally, the BiRNN-BiLSTM model carefully weighs the importance of different time points during the learning process, enhancing its ability to capture relative errors' characteristics and ability to discern actual trends. The BiLSTM model employs input, forget, and output gates to correct the signal transmission process, replacing ordinary nodes in the hidden layer of the RNN. This ensures that the gradient of the error function does not become too high during training, maintaining a strict time step for the gradient of the error function. The

Dropout mechanism is implemented to put some neurons into a failed state, preventing mutual adaptation during training.

Analyzing the case study results reveals frequent age and other labor-related violations, indicating a high construction safety risk. As China experiences a decline in its demographic dividend and an increase in the aging population, this issue is particularly pronounced in the labor employment sector. Research indicates that the average age of labor workers is approaching 45 years old, with regulations prohibiting workers over 55 (for men) and 50 (for women) from engaging in heavy physical work on construction sites. Despite these regulations, violations related to the age of workers persist, significantly increasing the safety risk on construction sites and hindering urban management efforts. While the three projects effectively control the age of workers, there are deficiencies in acquiring qualification certificates. Managers should further optimize the details of real-name list control, implement qualification checks for construction workers, and enhance on-site construction safety evaluations. Additionally, the objective collection of worker types facilitates targeted safety education and training activities by managers.

The implementation of the real-name management system for workers entails unified management of detailed information, including personnel details, skill information, and credit information. This approach significantly enhances the relevance of labor management, allowing for the early detection and elimination of cases involving illegal employment, laborers with a negative record, and those not in compliance with warehouse regulations. Moreover, it reinforces vocational education for labor workers, enhances training supervision and management, and elevates industry standards.

The real-name system for construction workers necessitates establishing a training system documenting entry and skills training for labor workers. Those who fail the entry training assessment are prohibited from entering the workplace. Furthermore, there is a need to facilitate sharing real-name construction worker data across projects. Currently, individual projects and enterprises utilize relatively closed real-name management systems, hindering the interchangeability and sharing of data. To address this, the government should establish standards for sharing real-name construction worker data, mandating that relevant enterprises and units have access to this information. This initiative aims to enhance project management levels and contribute to the development of smart cities. In modern construction safety risk assessment, although various safety impact assessment and risk analysis methods proposed by experts and scholars have yielded valuable insights, there remains a need to collect more accident risk factors from construction sites. This collection process validates safety risk assessment results through practical examples, correct analysis of decision-making outcomes, and reduces analysis biases.

# CONCLUSIONS AND LIMITATIONS

## Conclusions

In this study, we quantify construction safety capability using the AHP method, establish an evaluation system across multiple abstract dimensions, and conduct construction safety evaluations by combining BiRNN and BiLSTM models. Validation results demonstrate that

the proposed model effectively captures sudden and anomalous sequence changes through its bidirectional structure. It comprehensively captures the temporal characteristics of the construction process by considering historical and future information. The use of BiRNN and BiLSTM for construction safety evaluation enhances the handling and utilization of complex datasets, improving the robustness and generalization ability of the model across different construction scenarios. This reduces the likelihood of accidents and enhances emergency response efficiency, ensuring worker safety and minimizing potential losses.

## Limitations

Both the BiRNN and BiLSTM models face challenges with gradient vanishing and exploding, mainly when dealing with long sequences. Although BiLSTM incorporates gating mechanisms to address these issues, they can still affect the model's stability and predictive accuracy in real-world construction safety evaluations. This limitation hinders the effective processing of extensive time series data, potentially impacting the reliability of safety assessments.

Future research could focus on developing advanced gradient management techniques or incorporating alternative architectures that further mitigate gradient-related issues in BiRNN and BiLSTM models. Exploring variations such as attention mechanisms or transformer models might offer improved performance in handling long sequences. Moreover, research into optimizing the computational efficiency of BiLSTM models is needed to address the high resource requirements. Techniques such as model pruning, quantization, or hybrid architectures that balance accuracy and computational demands could be investigated to enhance the feasibility of real-time safety monitoring systems.

Addressing the challenge of data imbalance could involve integrating synthetic data generation methods, anomaly detection techniques, or cost-sensitive learning approaches to improve the model's ability to detect rare safety events. Future studies should explore strategies to better represent and learn from sparse safety incident data.

### Funding

The authors received no funding for this work.

### Competing Interests

The authors declare there are no competing interests.

### Author Contributions

- Ming Ge conceived and designed the experiments, performed the experiments, performed the computation work, prepared figures and/or tables, authored or reviewed drafts of the article, and approved the final draft.
- Yongbo Yuan performed the experiments, analyzed the data, performed the computation work, authored or reviewed drafts of the article, and approved the final draft.

## Data Availability

The code is available in the Supplementary File.

The data is available at Zenodo: Ashrant Aryal, Chao Wang, Chintan Vijay Vora, & Sueed A Willoughby. (2022). Development of Robotics & Automation Roadmap for Road Construction/Maintenance Projects [Data set]. Zenodo. https://doi.org/10.5281/zenodo.7930450.

## Supplemental Information

Supplemental information for this article can be found online at http://dx.doi.org/10.7717/peerj-cs.2351#supplemental-information.

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
