# Peer review of "Evaluation model design of project construction safety level based on bidirectional recurrent neural network (BiRNN) and bidirectional long short-term memory (BiLSTM)"

_PeerJ Computer Science, doi:10.7717/peerj-cs.2351_

## Round 0.1 · original submission · Major Revisions

Dear Authors,

Thank you for your submission to our esteemed journal. your manuscript has been carefully reviewed by the academic editor and independent experts in the field. Along with their feedback for improvement, please also consider my input for the improvement. Please carefully revise the manuscript in light of these suggestions and then resubmit.

Additional Editor Comments:

The paper introduces complex methodologies, such as the Analytic Hierarchy Process (AHP) and hybrid BiRNN-BiLSTM models, but the explanation of how these methods are integrated could be elaborated more.

it would be nice to contextualize the results more thoroughly. How do these results compare to existing methods or benchmarks?.

This research identifies 19 secondary causal factors, but the selection process for these factors is not fully explained. Please explain it in more detail

It would be good if more visual aids like flowcharts, diagrams, or tables could be added for more in depth discussion and presentation of the study.

The paper needs a through improvement in the language used.

thank you

Reviewer 1 ·

Basic reporting

I have carefully reviewed your manuscript titled "Evaluation Model Design of Project Construction Safety Level Based on BiRNN and BiLSTM" and would like to provide some constructive feedback to enhance the quality and clarity of your work.

 Your manuscript effectively outlines the capabilities of BiRNN and BiLSTM models in handling sequential data and capturing temporal dependencies. However, it is important to acknowledge the significant computational resources required by these models, particularly for large-scale projects. This computational demand can be a limiting factor for real-time safety evaluations where immediate feedback is essential. I recommend discussing potential strategies to mitigate these computational challenges, such as model optimization techniques or hardware acceleration.

 The introduction could benefit from a more comprehensive explanation of the BiRNN architecture. Emphasize its ability to capture bidirectional dependencies in sequential data and detail the complementary roles of the forward and backward passes in context capturing.

 The equations (Eqs. 5-7) should be seamlessly integrated into the text with clear explanations. Define the weight matrices and bias vectors, and elaborate on the role of each term in the equations. Ensure the transition from these equations to the final output sequence

Experimental design

See the 'Basic Reporting' for detailed comments

Validity of the findings

See the 'Basic Reporting' for detailed comments

Additional comments

See the 'Basic Reporting' for detailed comments

·

Basic reporting

The paper is technically well done with a good scientific sound.
The ensemble deep learning has been proposed must be explained in its philosophy as well. The horizontal combination among BiLSTM and BiRNN is interesting but it must showed to the audience in a better technical way. In my opinion this part has to be deeply revised

Experimental design

The experimental design has been well conducted. Anyway also this part should be revised considering my requests to the precedent step

Validity of the findings

the findings are really new in my opinion but the ensemble deep learning has been proposed must be reinforced and explained

Additional comments

None

·

Basic reporting

The paper titled Evaluation Model Design of Project Construction Safety Level Based on BiRNN and BiLSTM seems to be good but can be further improved by incorporating the following suggestions.

1. Please clearly elaborate the problem statement. Specify the exact challenges in current safety evaluation methods and how your proposed approach addresses these issues.

2. Justify the use of of AHP and the hybrid BiRNN-BiLSTM model , the methodology section could be expanded to provide a more detailed explanation of how these techniques are implemented.

3. Explain why BiRNN and BiLSTM were chosen over other deep learning models. Discuss their advantages in handling time-series data and capturing temporal dependencies, which are crucial for construction safety evaluation.
4. The results section could include more in-depth analysis and interpretation. Discuss what the MSE, RMSE, MAE, and MAPE values imply in the context of construction safety. Provide comparative analysis with other traditional methods to highlight the superiority of your approach.
5. The case study section should provide more context and details. Describe the construction project, the data collection process, and the specific scenarios analyzed.
6. Adding figures, tables, and charts to illustrate key points, such as the architecture of the hybrid model, the distribution of the secondary causal factors, and the comparison of performance metrics, would enhance the readability and impact of your paper.
7. A section on limitations and future research directions would be beneficial.

8. Please add more literature review by including more references to recent studies on deep learning applications in construction safety. Highlight how your work differs from and improves upon these existing studies.
9. Emphasize the practical implications of your findings for construction project managers and safety officers.
10. Please improve the language of your manuscript.
11. By addressing these points, your paper will provide a more comprehensive and compelling case for the use of deep learning in construction safety evaluation, offering valuable insights for both academia and industry professionals.

Experimental design

No Comments

Validity of the findings

No Comments

Additional comments

No Comments

---

## Round 0.2 · accepted · Accept

Thank you for your revisions, The relevant experts and I have now reviewed the revised article and we are happy to inform you that your manuscript is being recommended for publication. thank you for your contribution.

·

Basic reporting

All suggested modifications have been incorporated,

Experimental design

OK

Validity of the findings

OK

Additional comments

OK

Reviewer 4 ·

Basic reporting

Thank you for making efforts to address the raised comments.

Experimental design

No Comment

Validity of the findings

No Comment